

# Trophic niche but not abundance of Collembola and Oribatida changes with drought and farming system

Svenja Meyer[1], Dominika Kundel[2,3], Klaus Birkhofer[4], Andreas Fliessbach[3] and Stefan Scheu[1,5]

[1] Animal Ecology, J.F. Blumenbach Institute for Zoology and Anthropology, University of Göttingen, Göttingen, Germany
[2] Ecology, Department of Biology, University of Konstanz, Konstanz, Germany
[3] Department of Soil Sciences, Research Institute of Organic Agriculture (FiBL), Frick, Switzerland
[4] Department of Ecology, Brandenburg University of Technology, Cottbus, Germany
[5] Centre of Biodiversity and Sustainable Land Use, University of Göttingen, Göttingen, Germany

## ABSTRACT

Higher frequencies of summer droughts are predicted to change soil conditions in the future affecting soil fauna communities and their biotic interactions. In agroecosystems drought effects on soil biota may be modulated by different management practices that alter the availability of different food resources. Recent studies on the effect of drought on soil microarthropods focused on measures of abundance and diversity. We here additionally investigated shifts in trophic niches of Collembola and Oribatida as indicated by stable isotope analysis ($^{13}C$ and $^{15}N$). We simulated short-term summer drought by excluding 65% of the ambient precipitation in conventionally and organically managed winter wheat fields on the DOK trial in Switzerland. Stable isotope values suggest that plant litter and root exudates were the most important resources for Collembola (*Isotoma caerulea*, *Isotomurus maculatus* and *Orchesella villosa*) and older plant material and microorganisms for Oribatida (*Scheloribates laevigatus* and *Tectocepheus sarekensis*). Drought treatment and farming systems did not affect abundances of the studied species. However, isotope values of some species increased in organically managed fields indicating a higher proportion of microorganisms in their diet. Trophic niche size, a measure of both isotope values combined, decreased with drought and under organic farming in some species presumably due to favored use of plants as basal resource instead of algae and microorganisms. Overall, our results suggest that the flexible usage of resources may buffer effects of drought and management practices on the abundance of microarthropods in agricultural systems.

# INTRODUCTION

Soils in agricultural systems are not well buffered against changes in climate and therefore are likely to undergo profound transformations in face of future climate change. For Central Europe, the predicted climate changes include reduced precipitation during summer with consequent higher frequency of summer droughts (*EEA, 2017*; *Samaniego et al., 2018*). The

Corresponding author
Svenja Meyer, smeyer6@gwdg.de

resulting lower soil moisture levels are not only relevant for the water supply of crops, but also for soil biota and associated ecosystem functions, such as nutrient cycling and soil aggregate formation (*Geng et al., 2015*; *Spinoni, Naumann & Vogt, 2015*; *Zhang et al., 2018*). Studies on the effect of drought on soil microarthropods have focused on measures of diversity and abundance, and indicate either a negative (*Frampton, VandenBrink & Gould, 2000*; *Blankinship, Niklaus & Hungate, 2011*; *Kardol et al., 2011*; *Vestergård et al., 2015*) or no response (*Holmstrup et al., 2013*; *Krab et al., 2014*).

One way to better understand the response of soil microarthropods to increased drought conditions and to further connect them to processes such as nutrient cycling is to investigate changes in their feeding behavior. Soil organisms interact with each other in a food web with links of different strengths between the individual components (*Scheu, 1998*; *Hines et al., 2015*; *Potapov, Tiunov & Scheu, 2019*). From the perspective of soil microarthropods, some of these links may be particularly vulnerable to a decrease in soil moisture (*Barreto et al., 2021*). Bacteria and nematodes, for example, need water films between soil particles to move, grow and survive (*Erktan, Or & Scheu, 2020*), and thus are only available as a food resource for microarthropods at sufficiently high soil moisture levels. Further, drought may reduce plant biomass and thereby major basal resources of soil food webs such as leaf litter, roots and rhizodeposits (*Jaleel et al., 2009*; *Scheunemann et al., 2015*). Negative effects on these basal resources of soil food webs may cascade up to higher trophic levels resulting in decreased abundance and changes in trophic niches towards more drought-resistant resources like soil organic matter.

In agricultural systems trophic links are likely to vary with soil characteristics and management practices. The availability of resources for microarthropods, such as soil organic matter and microorganisms, differs substantially between management systems receiving organic or mineral fertilizers (*Mäder et al., 2002*; *Birkhofer et al., 2008*). Organically managed fields are characterized by higher abundances of microorganisms with a larger proportion of fungi compared to systems receiving only mineral fertilizer (*Haubert et al., 2009*). These form an important food resource for Collembola and Oribatida (*Schneider et al., 2004*; *Chahartaghi et al., 2005*; *Pollierer & Scheu, 2021*). Such differences in resource availability are likely to change the feeding behavior of microarthropods, which has been shown for several other compartments of soil food webs before (*Haubert et al., 2009*; *MacFadyen et al., 2009*; *Birkhofer et al., 2011*). However, the few studies that include Collembola and Oribatida species only documented subtle or no niche shifts with changes in environmental conditions (*Korotkevich et al., 2018*; *Krause et al., 2019*).

In this study we use stable isotope analysis to characterize trophic niches of abundant species of Collembola and Oribatida in replicated plots of long-term conventional and organic farming systems. The ratio of the stable isotopes of nitrogen allows insight into the trophic level of consumers due to enrichment in $^{15}$N in higher trophic levels, whereas the ratio of carbon stable isotopes reflects the utilization of basal resources (*Post, 2002*; *Potapov, Tiunov & Scheu, 2019*). In previous studies, stable isotope analysis has mainly been used to characterize the trophic structure of soil animal communities of different habitats and to generally clarify the usage of basal resources by certain taxonomic groups (*Scheu & Falca, 2000*; *Potapov, Tiunov & Scheu, 2019*). Only few studies employed stable

isotope analysis to investigate the response of the trophic structure of soil food webs to different experimental treatments such as different farming systems (*Haubert et al., 2009*; *Birkhofer et al., 2011*; *Susanti et al., 2021*) or track changes in trophic niches induced by changes in environmental conditions (*Birkhofer et al., 2016*; *Korotkevich et al., 2018*; *Krause et al., 2019*). Dry conditions were shown to increase $\delta^{15}N$ values of Oribatida in forests possibly due to trophic shifts resulting from changes in microbial activity and community composition (*Melguizo-Ruiz et al., 2017*). Further, drought is likely to increase periods of starvation due to lower availability of resources that depend on high soil moisture. Starvation has been shown to increase $\delta^{13}C$ and $\delta^{15}N$ values of the body tissue of animals in part as a result of metabolizing lipids, which are depleted in $^{13}C$ (*Adams & Sterner, 2000*; *Oelbermann & Scheu, 2002*; *Haubert et al., 2009*). Effects of drought on stable isotope ratios of microarthropods are likely to differ between different farming systems which comprise differently structured soil food webs (*Birkhofer et al., 2011*). Additionally, a reduction in soil moisture can be buffered in organically managed fields due to high soil organic carbon contents that result in more structured soils with a higher water holding capacity (*Lotter, Seidel & Liebhardt, 2003*; *Kundel et al., 2020*). However, the interactive effects of experimental drought and farming systems on the trophic behavior of microarthropods, to the best of our knowledge, have not been investigated before.

Besides looking into mean values of stable isotope ratios of carbon and nitrogen separately, we further include estimates of trophic niche sizes combining measurements of both isotopes in a two-dimensional space (*Bearhop et al., 2004*; *Jackson et al., 2011*). Niche sizes are proposed to become narrower in stable, deterministic environments due to more specialization (*Giller, 1996*). In line with this assumption empirical studies on trophic niches of soil animals suggest that trophic niche sizes are smaller in undisturbed compared to disturbed habitats (*Korotkevich et al., 2018*). We, therefore, expected disturbances such as drought to enlarge the trophic niche of Collembola and Oribatida. Likewise, regarding farming systems, we assumed that a conventional system based on the input of mineral fertilizer without organic fertilizers to represent a more disturbed system compared to an organically managed system receiving manure. Conditions in the conventional system should hence force consumers to enlarge their trophic niche.

Here, we investigated the trophic niches of individual species of Collembola and Oribatida as affected by experimental drought and conventional *versus* organic farming. We hypothesized (1) trophic niches to vary among species indicating the occupation of different trophic levels and the utilization of different basal resources with intraspecific differences between the conventional and the organic farming system. Further, we hypothesized (2) drought to change the trophic ecology of the studied microarthropod species expressed by increased $\delta^{13}C$ and $\delta^{15}N$ values of individual species with this being more pronounced in conventionally compared to organically managed fields. We further hypothesized that (3) trophic niche sizes are larger in the experimental drought treatments and the conventional farming system as more severely disturbed systems.

## MATERIALS & METHODS

### Study site

The study was performed in 2017 in the DOK trial, an agricultural long-term field experiment established in 1978 comparing different organic and conventional farming systems. The DOK trial is located in Therwil, Switzerland, at 300 m above sea level on a Haplic Luvisol on deep deposits of alluvial loess (*Fließbach et al., 2007*). The mean annual temperature over the last five years was 10.5 °C and the mean annual precipitation was 842 mm (*Krause et al., 2020*). For this study we used winter wheat fields with soybean as the previous crop. The experimental fields were organized in four blocks each comprising a conventionally and an organically managed field (factor farming system, CONMIN and BIODYN systems of the DOK trial, respectively). Conventionally managed fields received mineral fertilizer (40–60 kg N/ha in March, April and May), herbicides (0.1 l/ha of Husar OD, Bayer, Zollikofen, Switzerland, and 1 l/ha of Mondera, Switzerland, once in March), insecticides (0.1 l/ha of Audienz; Omya, Oftringen, Switzerland, in May) and fungicides (1.5 l/ha Pronto Plus in April and 1 l/ha AviatorXpro and Miros FL in May; all Bayer) as well as plant growth regulators (1.5 l/ha Cycocel extra; Omya, in March). Organically managed fields received only organic fertilizers (farmyard manure, compost and slurry), biodynamic preparations and mechanical weed control (*Krause et al., 2020*; *Kundel et al., 2020*). All fields were ploughed up to a depth of 20 cm and seedbed preparation was done with a tooth harrow to a depth of 10 cm. In both systems 415 grains/m$^2$ were sown. All fields followed the same 7-year crop rotation with soybean as the preceding crop. On each field one drought treatment and one control plot were established (factor drought). We simulated drought by using experimental rainout-shelters that excluded 65% of the precipitation (for details on the shelter construction see *Kundel et al., 2018*). On the control plots, we established a similar shelter construction with the difference, that it did not reduce precipitation entering the plot. Thereby, we accounted for possible side effects caused by the roof construction itself (*Kundel et al., 2018*).

### Sampling

Samples were taken in May, eight weeks after the establishment of the experiment, with soil cores of 5 and 20 cm diameter to a depth of 10 cm in the center of the plots ($n = 16$). Microarthropods with high densities, in our case the Collembola *Mesaphorura* sp. and Oribatida, were taken from the five cm cores. For the other Collembola, *i.e., Isotoma caerulea, Isotomurus maculatus* and *Orchesella villosa*, abundances in the small cores were too low to obtain enough material for stable isotope measurements, so we took them from the large soil cores that were initially taken to extract macrofauna. Soil animals were extracted from intact soil cores by gradually increasing the temperature from 25 to 55 °C over ten days (*Macfadyen, 1961*; *Kempson, Lloyd & Ghelardi, 1963*), collected into a glycol-water solution (1:1) in canisters underneath the soil coresand and stored in 70% ethanol. We first sorted the extracted animals to order level under a stereomicroscope (Stemi 2000; Zeiss). Thereafter, Collembola and Oribatida were identified to species or genus level under the microscope (Axioplan; Zeiss). As slide-mounting medium we used 70% ethanol, because other commonly used solutions like lactic acid may change stable

isotope compositions. We used keys by *Hopkin (2007)*, *Fjellberg (1998)*; *Fjellberg (2007)* and *Weigmann (2006)*.

## Stable isotope analysis

The four most abundant Collembola taxa (*I. caerulea*, *I. maculatus*, *O. villosa* and *Mesaphorura* sp.) and the two most abundant oribatid mite species (*Scheloribates laevigatus* and *Tectocepheus sarekensis*) were chosen for stable isotope analysis. To achieve at least 10 µg of animal dry weight per sample we used 1–14 individuals per sample. To have at least three values for every species x drought x farming system combination we included pseudoreplicates in plots with many individuals (Table 1). Animals were weighed into tin capsules and dried at 60 °C for 24 h. Wheat from every plot was dried, milled and weighed into tin capsules (ca. 1 mg per sample). Stable isotope analysis of animals was done with a coupled setup of an elemental analyzer (Eurovector, Milano, Italy) and a mass spectrometer (Delta Vplus, Thermo Fisher Scientific, Bremen, Germany) adjusted for small sample sizes (*Langel & Dyckmans, 2014*). Stable isotope analysis of wheat was done with another set of elemental analyzer and mass spectrometer (Flash 2000 elemental analyser coupled to a DELTA Plus XP continuous-flow IRMS *via* a ConFlo IV interface, Thermo Fisher Scientific, Bremen, Germany). Variations in stable isotope ratios including baseline correction were expressed using the delta notation with $\Delta X = (R_{SAMPLE}/R_{STANDARD})/R_{STANDARD} \times 1000$ with $X$ representing the target isotope ($^{13}$C, $^{15}$N), and $R_{SAMPLE}$ and $R_{STANDARD}$ the ratios of the heavy to the light isotope ($^{13}$C/$^{12}$C, $^{15}$N/$^{14}$N) of the sample and the standard, respectively. As standard for $^{13}$C PeeDee Belemnite and for $^{15}$N atmospheric air was used (*Coplen et al., 2002*). Acetanilide was used for internal calibration.

## Statistical analyses

All statistical analyses were done in R version 4.0.2 (*R Development Core Team, 2020*).

We calculated mean abundances for each species. Abundance data were analyzed with linear mixed effects models (LMMs) for individual species with farming system and drought as fixed factors, and field as random factor using the package nlme (*Pinheiro et al., 2021*).

Stable isotope data were baseline corrected using wheat stable isotope values of the respective plot and analyzed with a LMM with farming system and drought as fixed factors, and plot as random factor to account for differences in sample size. Because the interaction species × drought as well as species × farming system was significant (Table 2), we ran individual LMMs for each species to detect species-specific effects of drought and farming system. In these models we again included drought and farming system and their interaction as fixed factors, and plot as random factor.

The size of the isotopic niches of each species in the two farming systems and in the two drought treatments was calculated and visualized with the R package SIBER (*Jackson et al., 2011*). Standard ellipse areas with a correction for small sample sizes (SEAc) based on maximum likelihood were estimated and used to visualize isotopic niches of all species in the two farming systems and drought treatments. To compare isotopic niche widths between farming systems and drought treatments within species, Bayesian multivariate normal distributions were fitted to the two levels of the factor farming

**Table 1  Number of replicates for stable isotope measurements.** Per farming system (conventional, conv; organic, org) and drought treatment (control, roof).

| Species | Farming system | Drought | Number of replicates | |
|---|---|---|---|---|
| | | | $\delta^{13}$C | $\delta^{15}$N |
| Isotoma caerulea | conv | control | 5 | 5 |
| | | roof | 4 | 3 |
| | org | control | 6 | 6 |
| | | roof | 4 | 3 |
| Isotomurus maculatus | conv | control | 4 | 5 |
| | | roof | 4 | 4 |
| | org | control | 5 | 5 |
| | | roof | 4 | 3 |
| Orchesella villosa | conv | control | 6 | 7 |
| | | roof | 7 | 7 |
| | org | control | 6 | 6 |
| | | roof | 5 | 6 |
| Scheloribates laevigatus | conv | control | 4 | 4 |
| | | roof | 6 | 6 |
| | org | control | 7 | 6 |
| | | roof | 7 | 7 |
| Tectocepheus sarekensis | conv | control | 6 | 6 |
| | | roof | 6 | 5 |
| | org | control | 5 | 5 |
| | | roof | 4 | 4 |

**Table 2  Results of LMMs on the effects of drought and farming system on the abundance, $\Delta^{13}$C and $\Delta^{15}$N values of abundant species of the mesofauna.** Significant effects are given in bold.

| | $\Delta^{15}$N | | | $\Delta^{13}$C | | |
|---|---|---|---|---|---|---|
| | df | F | P | df | F | P |
| Drought (D) | 1,12 | 2.74 | 0.124 | 1,12 | 3.09 | 0.104 |
| Farming system (F) | **1,12** | **13.84** | **0.003** | 1,12 | 3.02 | 0.108 |
| Species (S) | **4,71** | **32.56** | **<0.001** | **5,83** | **43.02** | **<0.001** |
| D x F | 1,12 | 2.19 | 0.164 | 1,12 | 0.04 | 0.844 |
| D x S | 4,71 | 0.72 | 0.58 | **5,83** | **2.92** | **0.018** |
| F x S | **4,71** | **10.11** | **<0.001** | **5,83** | **4.30** | **0.002** |
| D x F x S | 4,71 | 2.39 | 0.059 | 5,83 | 1.49 | 0.203 |

system and drought, with prior settings of length, number and iterations of sampling chains, and distribution parameters as recommended by *Jackson (2019)*. Based on these probability distributions Bayesian standard ellipse areas were calculated and plotted using the function siberDensityPlot() including 50%, 75% and 95% credible intervals. For statistical comparison of isotopic niche sizes of the farming systems and the drought treatments for individual species, we compared probability distributions from the Bayesian standard ellipses with 95% credible intervals.

## RESULTS

### Soil characteristics

Water holding capacity, pH and total carbon were higher in organically compared to conventionally managed fields (Table S1). Total carbon at our study site is equivalent to total organic carbon, because the soil is free of carbonates. Soil water content was decreased by experimental drought by 4.23% and was generally higher in organically compared to conventionally managed fields (Fig. S1; see *Meyer et al., 2021*).

### Abundance

Based on their mean abundance the six mesofauna taxa could be separated into two groups of high and low abundance with abundances of the former being 23 to 73 times higher than that of the latter. Highly abundant taxa included *S. laevigatus, T. sarekensis* and *Mesaphorura* sp. (overall average of 7648 ± 1528, 7392 ± 1286 and 5312 ± 1734 ind m$^{-2}$, respectively; mean ± SE). Species with low abundances included *I. caerulea, I. maculatus* and *O. villosa* (106.8 ± 32.7, 105.0 ± 29.3 and 227.5 ± 49.8 ind m$^{-2}$, resepectively). Generally, abundances of individual species did not change significantly with drought treatment or farming system (Table 3, Fig. S2).

### Isotope values

Mean stable isotope values were significantly different between species, spanning over two δ units for $^{13}$C and over four δ units for $^{15}$N (Fig. 1, Table 2). The $\Delta^{13}$C values of the two oribatid mite species were three to four δ units higher than those of the three Collembola species. Mean $\Delta^{15}$N values spanned over four δ units with the values of *S. laevigatus* exceeding those of the other species by three to four δ units.

The $\Delta^{13}$C but not $\Delta^{15}$N values differed significantly among the studied mesofauna species between the drought treatments (Table 2), with this pattern being driven by a significant reduction in the $\Delta^{13}$C values of *S. laevigatus* under drought; $\Delta^{13}$C values of the other species were not significantly affected by drought (Fig. 2, Table 3). By contrast, both $\Delta^{13}$C and $\Delta^{15}$N values of mesofauna species varied significantly with farming system (significant species × farming system interaction; Table 2). In organically managed fields the $\Delta^{13}$C value of *T. sarekensis* and the $\Delta^{15}$N values of *I. caerulea* and *O. villosa* significantly exceeded those in conventionally managed fields (Fig. 3, Table 3).

Drought significantly reduced the isotopic niche width of *S. laevigatus* ($P = 0.016$), *I. caerulea* ($P = 0.003$) and *I. maculatus* ($P = 0.032$) (Fig. 4), with isotopic niches of *S. laevigatus* partly overlapping between the two drought treatments, whereas in *I. caerulea* and *I. maculatus* they overlapped in full (Fig. 5). Further, the isotopic niche space of *I. caerulea* and *I. maculatus* was significantly smaller in organically compared to conventionally managed fields, with isotopic niches of *I. caerulea* partly overlapping between the two farming systems, whereas those of *I. maculatus* overlapped in full (Fig. 6).

## DISCUSSION

The species studied were selected based on two criteria: sufficiently high abundance combined with sufficiently high biomass for stable isotope analyses, and therefore can be

**Table 3  Results of LMM on the effects of drought, farming system and species identity on the abundance, $\Delta^{13}$C and $\Delta^{15}$N values of the studied mesofauna species.** Significant effects ($P < 0.05$) are given in bold.

| | *Scheloribates laevigatus* | | | *Tectocepheus sarekensis* | | | *Isotoma caerulea* | | | *Isotomurus maculatus* | | | *Orchesella villosa* | | | *Mesaphorura sp.* | | |
|---|---|---|---|---|---|---|---|---|---|---|---|---|---|---|---|---|---|---|
| | df | F | P | df | F | P | df | F | P | df | F | P | df | F | P | df | F | P |
| **Abundance** | | | | | | | | | | | | | | | | | | |
| Drought (D) | 1,6 | 0.01 | 0.943 | 1,6 | 2.68 | 0.153 | 1,6 | 0.88 | 0.384 | 1,6 | 3.06 | 0.131 | 1,6 | 1.05 | 0.344 | 1,6 | 0.53 | 0.495 |
| Farming system (F) | 1,6 | 1.80 | 0.228 | 1,6 | 0.01 | 0.910 | 1,6 | 0.83 | 0.396 | 1,6 | 0.13 | 0.728 | 1,6 | 1.63 | 0.249 | 1,6 | 1.79 | 0.229 |
| D x F | 1,6 | 0.84 | 0.394 | 1,6 | 0.42 | 0.540 | 1,6 | 1.51 | 0.266 | 1,6 | 0.49 | 0.510 | 1,6 | 0.09 | 0.779 | 1,6 | 4.56 | 0.077 |
| **d13C** | | | | | | | | | | | | | | | | | | |
| Drought (D) | **1,6** | **17.08** | **0.001** | 1,6 | 1.16 | 0.310 | 1,6 | 0.54 | 0.485 | 1,6 | 2.51 | 0.157 | 1,6 | 0.01 | 0.922 | 1,6 | 0.10 | 0.766 |
| Farming system (F) | 1,6 | <0.01 | 0.960 | **1,6** | **14.11** | **0.005** | 1,6 | 1.16 | 0.312 | 1,6 | 0.88 | 0.379 | 1,6 | 0.36 | 0.562 | 1,6 | 0.43 | 0.536 |
| D x F | 1,6 | 0.07 | 0.798 | 1,6 | 0.04 | 0.851 | 1,6 | 0.19 | 0.676 | 1,6 | 2.68 | 0.145 | 1,6 | 0.13 | 0.726 | 1,6 | 0.75 | 0.419 |
| **d15N** | | | | | | | | | | | | | | | | | | |
| Drought (D) | 1,6 | 0.03 | 0.867 | 1,6 | 0.04 | 0.845 | 1,6 | 0.10 | 0.759 | 1,6 | 0.21 | 0.663 | 1,6 | 1.44 | 0.261 | – | – | – |
| Farming system (F) | 1,6 | 4.30 | 0.062 | 1,6 | 2.82 | 0.132 | **1,6** | **15.47** | **0.008** | 1,6 | 2.69 | 0.145 | **1,6** | **15.90** | **0.003** | – | – | – |
| D x F | 1,6 | 0.09 | 0.771 | 1,6 | 0.22 | 0.649 | 1,6 | 3.92 | 0.095 | 1,6 | 2.22 | 0.180 | 1,6 | 0.04 | 0.855 | – | – | – |

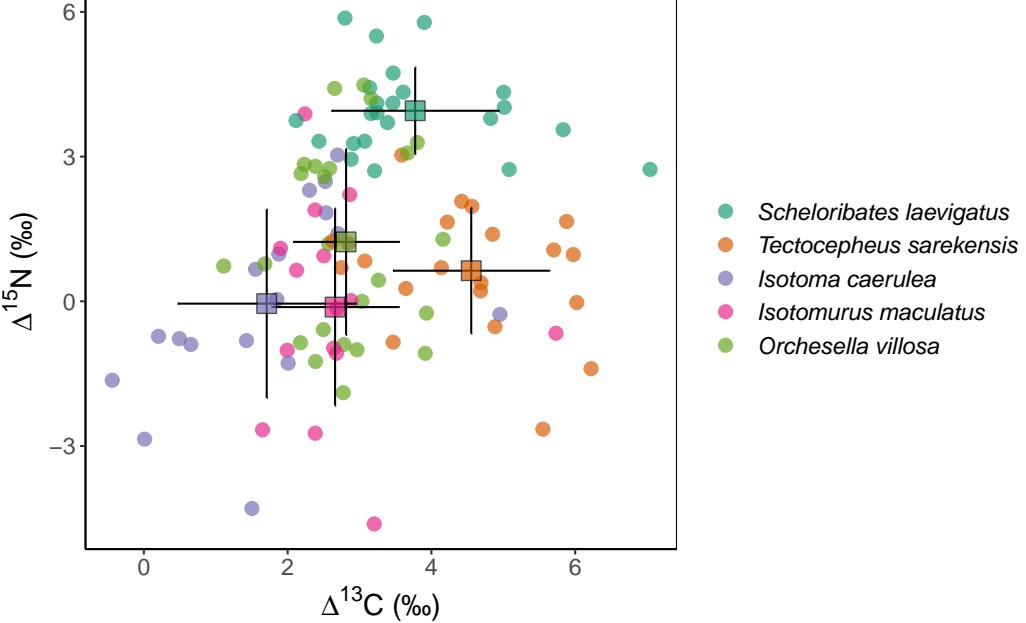

**Figure 1  Stable isotope values of Collembola and Oribatida.** Mean (±standard deviation) $\Delta^{13}$C and $\Delta^{15}$N values of two species of Oribatida (*Scheloribates laevigatus*, *Tectocepheus sarekensis*) and three species of Collembola (*Isotoma viridis*, *Isotomurus maculatus*, *Orchesella villosa*); data are calibrated against stable isotope values of wheat in the respective plot.

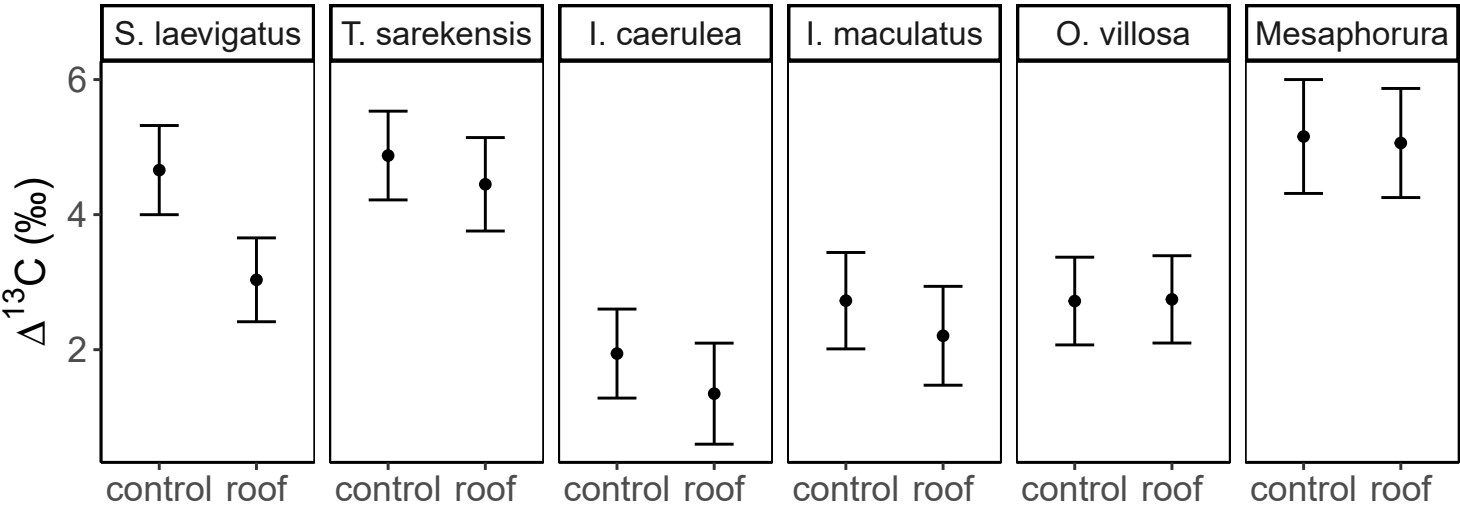

**Figure 2** **Stable isotope values of Collembola and Oribatida in drought and control treatment.** Mean (±95% confidence interval) $\Delta^{13}$C values in control and drought treatments (roof) for two species of Oribatida (*Scheloribates laevigatus*, *Tectocepheus sarekensis*) and four Collembola taxa (*Isotoma caerulea, Isotomurus maculatus, Orchesella villosa, Mesaphorura* sp.); for statistical analysis see Table 3.

considered as the most important Collembola and Oribatida species of the system regarding energy flux and functioning. Interestingly, the farming system and the experimental drought did not affect the abundances of the studied species significantly, but affected their trophic niches as indicated by stable isotope analyses.

### Trophic positions

Overall, stable isotope values of the studied microarthropods spanned two $\delta$ units in $^{13}$C and four $\delta$ units in $^{15}$N, indicating the utilization of different C resources and the representation of at least two trophic levels, assuming an enrichment of about 3 $\delta$ units per trophic level (*Post, 2002*). Based on the $\Delta^{13}$C and $\Delta^{15}$N values of the individual species, the studied taxa can be separated into three groups, the three Collembola species, the Oribatida species *T. sarekensis* and the Oribatida species *S. laevigatus*.

The three Collembola species *I. caerulea, I. maculatus* and *O. villosa* had $\Delta^{15}$N values close to zero, indicating they are closely linked to wheat plants and suggesting that they live as primary decomposers that are little enriched in $^{15}$N (−0.05‰, −0.12‰, 1.23‰, respectively). Earlier studies also found large epi- and hemiedaphic Collembola species, such as the ones we studied, to predominantly feed on plant-derived resources in both agroecosystems and forests (*Pollierer et al., 2009*; *Birkhofer et al., 2016*; *Potapov et al., 2016*). *Ngosong et al. (2009)* further found plant rather than fungal resources to be incorporated by Collembola in agricultural systems, and results of the study of *Li et al. (2020)* suggest that root-derived carbon is a major resource.

Isotope values of $^{13}$C of both Oribatida species exceeded those of the three Collembola species by one to two $\delta$ units, indicating that both are linked to resources enriched in $^{13}$C. However, their $\Delta^{15}$N values indicated that they occupy different trophic levels with *T. sarekensis* living as primary decomposer and *S. laevigatus* as secondary decomposer or

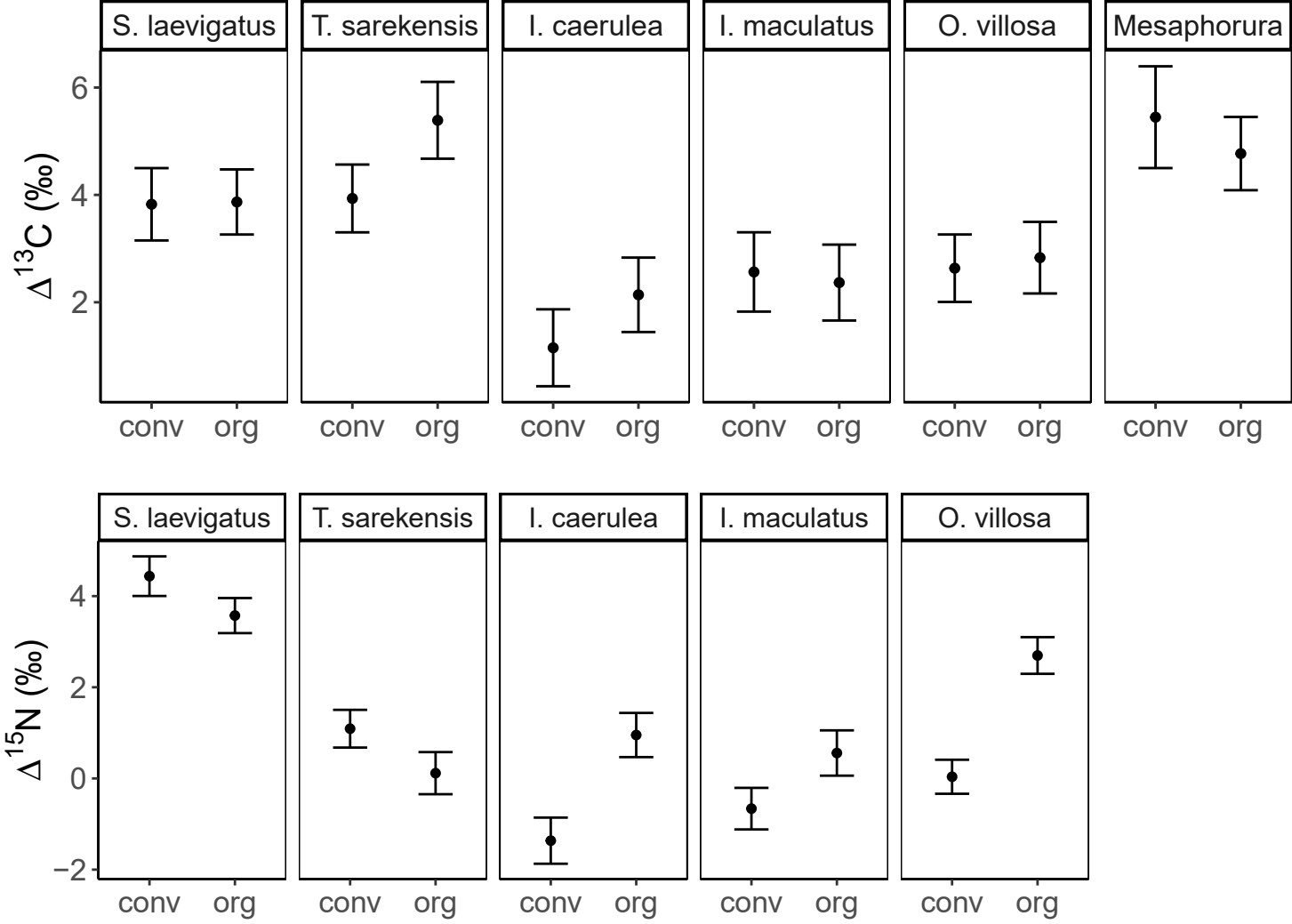

**Figure 3** **Stable isotope values of Collembola and Oribatida in two farming systems.** Mean (±95% confidence interval) $\Delta^{13}$C (upper panel) and $\Delta$15N values (lower panel) of two oribatid mite (*Scheloribates laevigatus, Tectocepheus sarekensis*) and four Collembola species (*Isotoma caerulea, Isotomurus maculatus, Orchesella villosa, Mesaphorura* sp.) in conventional (conv) and organic (org) farming systems; note that for *Mesaphorura* sp. only $\Delta^{13}$C values are shown. For statistical analysis see Table 3.

predator, similar to what has been previously suggested (*Schneider et al., 2004*; *Haynert et al., 2017*). The average $\delta^{13}$C value of *T. sarekensis* being 4.55‰ higher than plant litter indicates that *T. sarekensis* is linked to older carbon resources, probably soil organic matter in deeper soil layers (*Potapov, Tiunov & Scheu, 2019*). The average $\delta^{15}$N value of *S. laevigatus* being 3.77‰ higher than that of plant litter indicates a mixed diet consisting of mainly microorganisms, but in part also microbial feeders such as nematodes.

### Farming system

Our second hypothesis was partly supported by the significantly higher isotope values of *T. sarekensis*, *I. caerulea* and *O. villosa* in the organic compared to the conventional farming system. However, we did not find differences for the other taxa, which is in line with

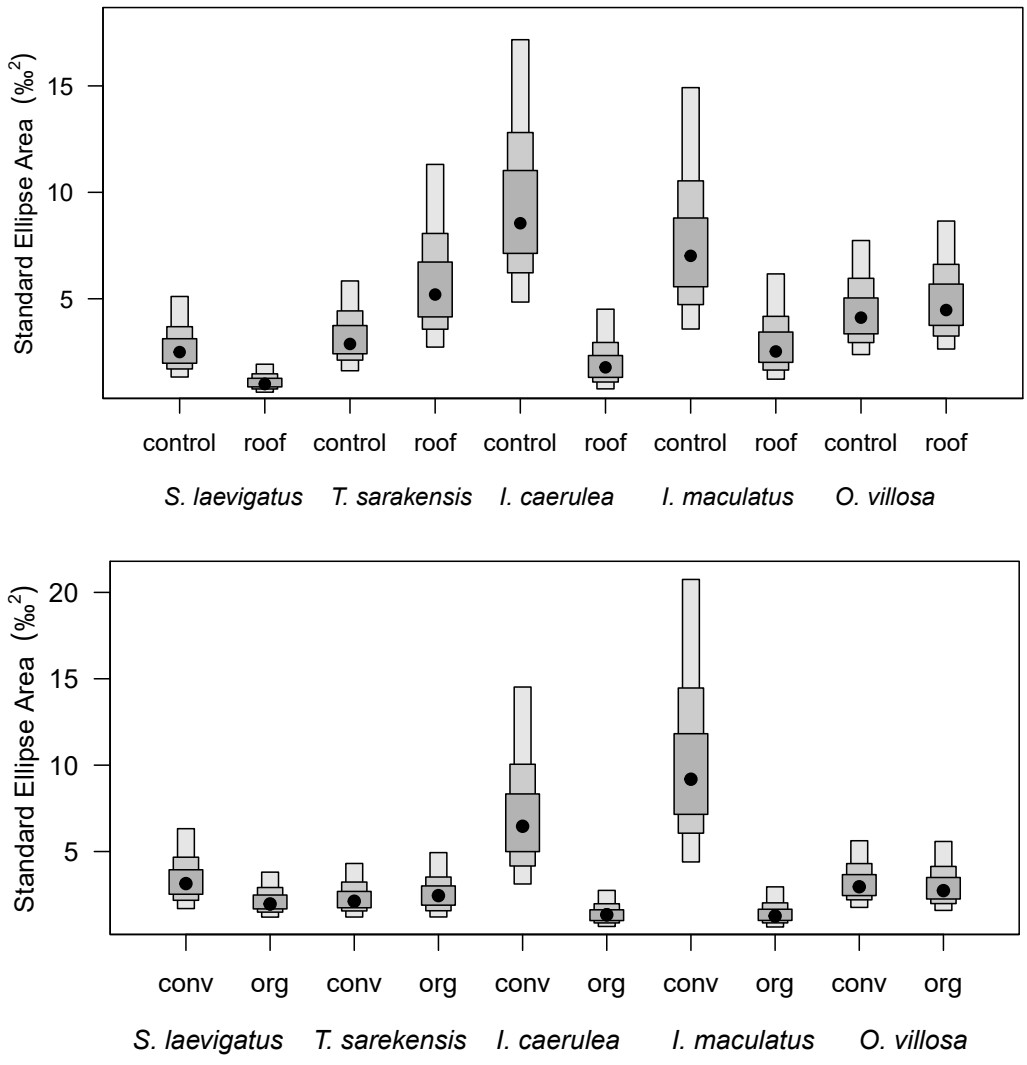

**Figure 4  Isotopic niche sizes in drought and control plots and in two different farming systems.** Probability distribution of the sizes of isotopic niches of five microarthropod species (*Scheloribates laevigatus, Tectocepheus sarekensis, Isotoma caerulea, Isotomurus maculatus, Orchesella villosa*) in the drought (roof) and the control treatment (upper panel), and in conventional (conv) and organic (org) farming systems (lower panel). Points show posterior estimates of the Bayesian standard ellipse area with 50%, 75% and 95% credible intervals (from dark to light gray).

earlier studies comparing different agricultural systems (*Haubert et al., 2009*; *Birkhofer et al., 2011*; *Lagerlöf, Maribie & John, 2017*). The higher isotope values of *T. sarekensis, I. caerulea* and *O. villosa* in the organic farming system likely are related to the higher soil organic carbon content that was found in this system. The higher $\Delta^{13}$C values of *T. sarekensis* in organically compared to conventionally managed fields indicate that they more intensively feed on old carbon resources in the organic system, which is richer in soil organic matter due to long-term input of farmyard manure and compost (*Mäder et al., 2002*). In *I. caerulea* and *O. villosa* $\Delta^{15}$N values were higher in organically compared

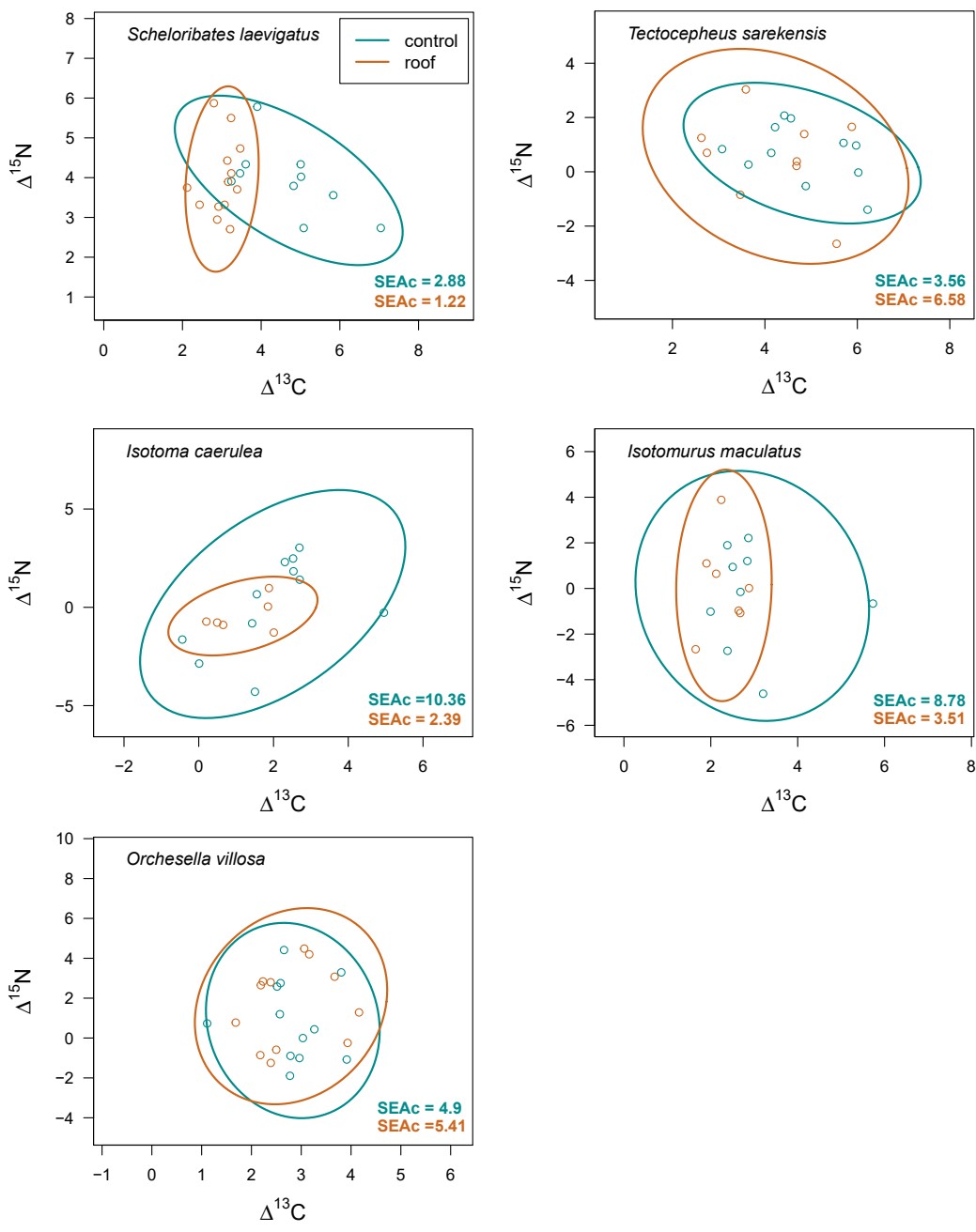

**Figure 5** **Isotopic niche space of Collembola and Oribatida in drought and control treatment.** Isotopic niche space of two oribatid mite (*Scheloribates laevigatus*, *Tectocepheus sarekensis*) and three Collembola species (*Isotoma caerulea*, *Isotomurus maculatus*, *Orchesella villosa*) in the drought (roof, orange) and the control (turquoise) treatment. Standardized ellipses (SEAc) account for different sample sizes between taxa and small sample sizes per taxon and encompass approximately 95% of the data; see Methods.

to conventionally managed fields, pointing to a higher proportion of microorganisms in their diet in the organic system. In fact, previous studies conducted in the same long-term experiment as the present study found higher microbial biomass in the organically than
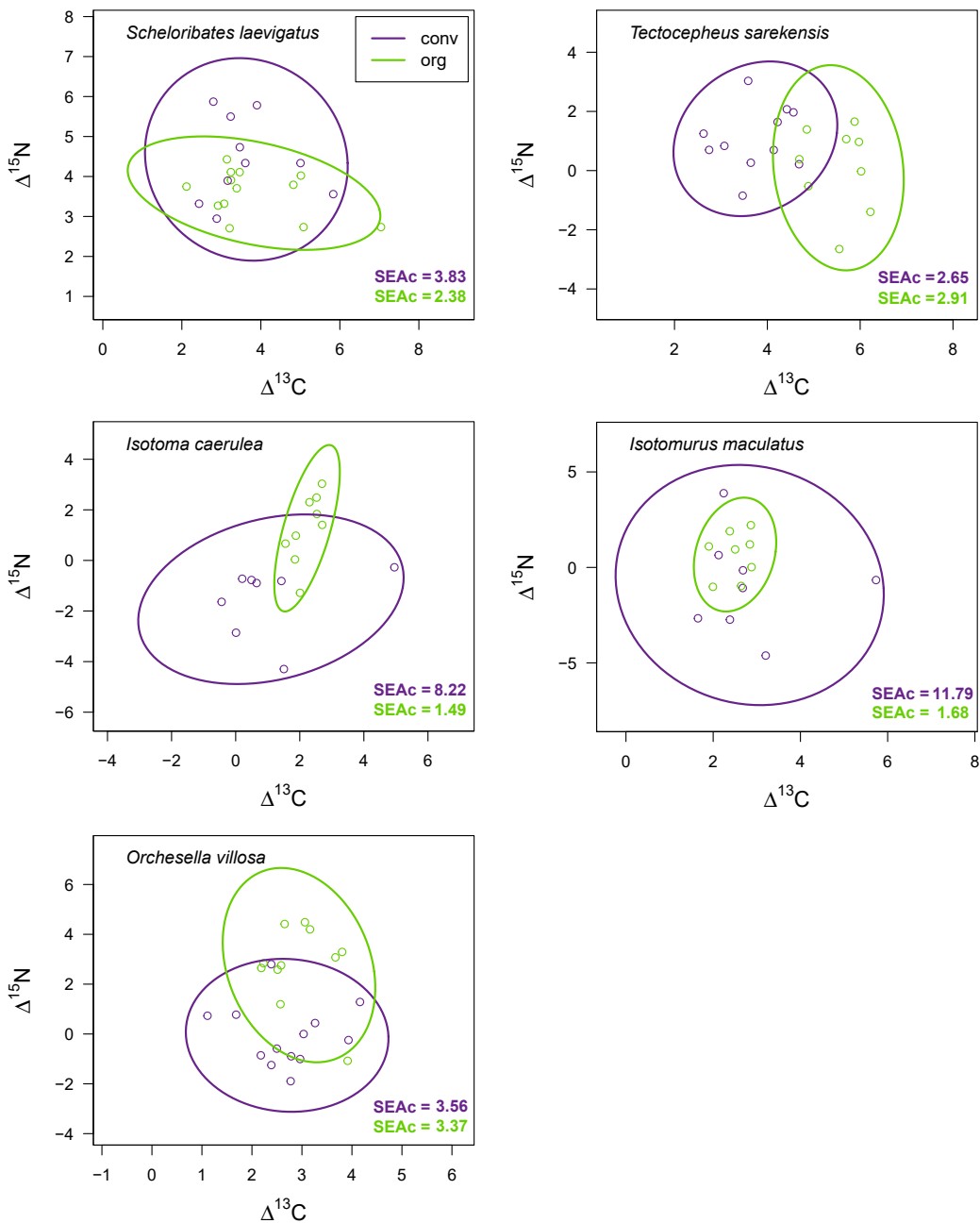

**Figure 6** **Isotopic niche space of Collembola and Oribatida in two different farming systems.** Isotopic niche space of two oribatid mite (*Scheloribates laevigatus*, *Tectocepheus sarekensis*) and three Collembola species (*Isotoma caerulea, Isotomurus maculatus, Orchesella villosa*) in conventionally (conv, purple) and organically (org, green) managed fields. Standardized ellipses (SEAc) account for different sample sizes between taxa and small sample sizes per taxon and encompass approximately 95% of the data; see Methods.

the conventionally managed fields indicating a higher availability of microbes as food resource (*Esperschütz et al., 2007*; *Fließbach et al., 2007*). This is likely to be a consequence of higher soil water content and soil organic carbon in the organically managed fields. This

effect might further be enhanced by higher $\delta^{15}$N values of the organic fertilizer (farmyard manure) compared to the inorganic fertilizer in the conventional system (*Birkhofer et al., 2011*). Higher stable isotope values in the organic system may additionally be caused by stable isotope enrichment of soil organic matter due to stronger internal nutrient cycling (*Vervaet et al., 2002*; *Hobbie & Ouimette, 2009*).

Besides the comparison of mean stable isotope values, additional information on trophic shifts of species can be obtained by comparing trophic niche width and trophic niche space (*Behan-Pelletier, 1999*; *Bearhop et al., 2004*). Our hypothesis on changes in trophic niche width with farming system was based on the assumption that in less disturbed habitats consumers would have a greater range of potentially available food resources, from which species could select according to their preferences. By contrast, in more severely disturbed systems, preferred resources may not be available, forcing consumers to feed on a wider range of resources resulting in broader trophic niches. Our data support this hypothesis only partly for the two farming systems and, interestingly, in some species showed the opposite pattern for the drought treatment (see below). Variations in $\Delta^{13}$C values in *I. caerulea* and *I. maculatus* were small in the organic farming system indicating a diet consisting of fresh litter or root exudates, whereas in the conventional farming system diets varied more widely. This suggests the utilization of a wider range of resources including old litter and microorganisms resulting in increased $\delta^{13}$C values (*Potapov, Tiunov & Scheu, 2019*) or algae resulting in decreased $\delta^{13}$C values (*Tozer et al., 2005*). In the conventional system, the amount of litter input is low and limited to plant residues from the crop plant, *i.e.,* mainly roots, whereas in the organic system plant residues in the organic fertilizer provide additional food resources. Further, the amount of rhizodeposits in the conventional system is likely to be lower than in the organic system, thereby providing fewer resources to the belowground food web (*Jones et al., 2001*; *Li et al., 2016*; *Wang, Chapman & Yao, 2016*). The lower availability of preferred food resources in the conventional compared to the organic farming system may force soil invertebrates to broaden their trophic niche. Further, in the conventional farming system more algae may be present due to the scarcity of weeds (*Meyer et al., 2021*) providing additional food resources that are not equally available in organic farming systems.

## Drought

Drought significantly decreased soil moisture in both farming systems. However, contradicting our second hypothesis, drought did not affect stable isotope values of most taxa and there was no significant interaction with farming system. Only *S. laevigatus* had lower $\Delta^{13}$C and constantly high $\Delta^{15}$N values in the drought treatment indicating prey switching. Assuming that *S. laevigatus,* at least in part, feeds on nematodes, this might represent a switch from microbial-feeding to plant-feeding nematodes, due to microbial-feeding nematodes being heavily stressed under dry conditions due to reduced microbial activity (*Kundel et al., 2020*). In contrast to conventional farming and contrasting our second hypothesis, drought decreased the trophic niche width in some species (*S. laevigatus, I. caerulea* and *I. maculatus*). For *S. laevigatus* this was caused by lower $\Delta^{13}$C values in the drought treatment, probably due to prey switching (see above). In the two

Collembola species the decreased trophic niche width was due to decreased variation in $\Delta^{13}$C values, but not lower mean $\Delta^{13}$C values, indicating more restricted consumption of plant-derived resources rather than algae and microorganisms. Accessibility of algae and microorganisms is likely to decrease at low soil moisture, whereas the availability of (higher) plant-derived resources may be less affected. In fact, plant-related variables, including root biomass, shoot biomass and grain yield, did not differ between the drought treatments in this experiment (*Kundel et al., 2020*). For *I. caerulea*, additionally, the smaller variation in $\Delta^{15}$N values, but no changes in mean $\Delta^{15}$N values, in the drought treatment supports the conclusion of narrower trophic niches due to more pronounced feeding on plant material.

## CONCLUSIONS

Drought did not significantly affect mean stable isotope values of most of the studied mesofauna species, but trophic niche width and space changed significantly, highlighting the relevance of these trophic niche characteristics for tracking effects of changes in environmental factors on soil food webs. Our results provide further evidence that in agricultural fields both plant litter and root-derived carbon play an important role as food resource for soil microarthropods. Overall, our data indicate that short-term drought as well as organic farming reduces the diversity of the resources used by soil microarthropods and favors the use of plants as basal resource for Collembola and Oribatida instead of microorganisms and algae. At the same time, the abundances of Collembola and Oribatida were not affected, suggesting that a flexible usage of resources may buffer negative effects of drought conditions on microarthropod communities in agricultural fields.

## ACKNOWLEDGEMENTS

We thank Guido Humpert for help with field work and Barbara Wozniak for help with sorting of animals. We further thank the Physiological Plant Ecology Group led by Ansgar Kahmen at the University of Basel for the measurement of stable isotopes in wheat plants.

### Funding

The study was funded in the framework of the 2015-2016 BiodivERsA COFUND call with the national funders the German Research Foundation (DFG), the Swiss National Science Foundation (SNSF), the Swedish Research Council (Formas), the Ministry of Economy and Competitiveness (MINECO) and Estonian Research Council (ETAG). The DOK trial is funded through the Swiss Federal Office of Agriculture. We received support from the Open Access Publication Funds of the University Göttingen. The funders had no role in study design, data collection and analysis, decision to publish, or preparation of the manuscript.

## Grant Disclosures

The following grant information was disclosed by the authors:
The German Research Foundation (DFG).
The Swiss National Science Foundation (SNSF).
The Swedish Research Council (Formas).
The Ministry of Economy and Competitiveness (MINECO).
Estonian Research Council (ETAG).
Swiss Federal Office of Agriculture.
University Göttingen.

## Competing Interests

The authors declare there are no competing interests.

## Author Contributions

- Svenja Meyer performed the experiments, analyzed the data, prepared figures and/or tables, authored or reviewed drafts of the paper, and approved the final draft.
- Dominika Kundel conceived and designed the experiments, performed the experiments, authored or reviewed drafts of the paper, and approved the final draft.
- Klaus Birkhofer and Stefan Scheu conceived and designed the experiments, analyzed the data, authored or reviewed drafts of the paper, and approved the final draft.
- Andreas Fliessbach conceived and designed the experiments, authored or reviewed drafts of the paper, and approved the final draft.

## Data Availability

The raw measurments of abundance and stable isotope values of Collembola, Oribatida and wheat plants are available in the Supplementary File.

## Supplemental Information

Supplemental information for this article can be found online at http://dx.doi.org/10.7717/peerj.12777#supplemental-information.

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
