# Peer review of "Trophic niche but not abundance of Collembola and Oribatida changes with drought and farming system"

_PeerJ, doi:10.7717/peerj.12777_

## Round 0.1 · original submission · Minor Revisions

Thank you for your manuscript. I found it well written and of considerable interest. Your manuscript was reviewed by two external reviewers. Both suggest some minor revisions for clarity. I would appreciate more data on the cropping system (e.g. soil type, tillage regime, etc) being included if possible. Reviewer 2 also indicates this data would help to validate some of the discussion/conclusion points, and I agree.

Reviewer 1 ·

Basic reporting

no comment

Experimental design

Please see pdf as I had questions about your design.

Validity of the findings

no comment

Additional comments

Dear author,

Thank you for your contribution. I enjoyed reading your manuscript, and hope my comments are useful to improve it.

Cheers

Annotated reviews are not available for download in order to protect the identity of reviewers who chose to remain anonymous.

·

Basic reporting

The writing is mostly clear (see attached PDF for comments on minor ambiguities). Professional English used throughout.

Literature and background appear sufficient.

Article structure, figures and tables appear professional. It doesn't appear that there is a colour scheme across the figures, in case that matters. Raw data on arthropod abundances and stable isotopes is shared.

There are statements in the discussion that are difficult to determine if they are self-contained or referring to the included references. For example, Line 257: "In fact, previous studies on the same experimental fields found higher microbial biomass in the organically than the conventionally managed fields indicating a higher availability of microbes as food resource, and this is likely to be a consequence of higher amounts of soil organic matter (Espersch├╝tz et al. 2007; Flie├čbach et al. 2007)." It's not fully clear if this study is hypothesizing that there is higher OM, or if the two referenced studies showed higher OM in these experimental fields, or if the two referenced studies found higher OM in organically managed fields elsewhere. See comments in discussion for other sentences that aren't clear. However, more importantly, these statements could be made stronger and more clear by including more information on the soil characteristics of the sampled soil in this experiment.

Experimental design

This study is original primary research, with well defined questions and hypotheses. The introduction outlines how the research fills an identified knowledge gap.

This study is part of a long term project, which appears to be well set-up for answering the questions at hand.

Experimental design (plot design) appears sufficient.

More information is needed on some key aspects of experimental design including field management (e.g. tillage regimes, cover cropping, etc), sampling design within the plots (e.g. how many samples per plot and where within the plot) and specimen preparation. In particular, tillage and cover cropping can have a large impact on soil moisture content, water infiltration, and water holding capacity.

It is curious that there is no data reported on the soil profile within the plots for this study, especially soil moisture content, percent organic matter, total carbon, total nitrogen (also pH and salinity could be interesting). What was the rainfall that year? These link with the questions at hand and would be highly beneficial to report and correlate.

Validity of the findings

The results appear valid.

Again, I am finding it difficult to evaluate the validity and/or strength of some of the discussion and conclusions without the provision of data on the soil profile, specifically soil moisture and organic matter content in plots. The statements can be made (as they are now, directing us to other references) but would be a lot stronger if the authors provided and assessed their own data.

Additional comments

I found the article to be very well written and interesting, and is of relevance with the continued movement towards sustainable agricultural practices. It is also refreshing to see work in agricultural systems on soil invertebrates, and particularly on research that attempts to tie together the workings of the soil food web. Nice work!

Please see attached pdf for in-text comments.

---

## Round 0.2 · accepted · Accept

Thank you for addressing all inquires and comments in the previous version. I am recommending your manuscript for publication.